# New Therapeutics in Alzheimer's Disease Longitudinal Cohort study (NTAD): study protocol

Juliette Helene Lanskey [ORCID],[1] Ece Kocagoncu [ORCID],[2] Andrew J Quinn [ORCID],[3] Yun-Ju Cheng,[4] Melek Karadag,[2] Jemma Pitt [ORCID],[3] Stephen Lowe,[5] Michael Perkinton,[6] Vanessa Raymont [ORCID],[7] Krish D Singh [ORCID],[8] Mark Woolrich [ORCID],[3] Anna C Nobre [ORCID],[3] Richard N Henson [ORCID],[1,9] James B Rowe [ORCID],[1,2] on behalf of the NTAD study group

For numbered affiliations see end of article.

**Correspondence to**
Juliette Helene Lanskey;
juliette.lanskey@mrc-cbu.cam.ac.uk

## ABSTRACT

**Introduction** With the pressing need to develop treatments that slow or stop the progression of Alzheimer's disease, new tools are needed to reduce clinical trial duration and validate new targets for human therapeutics. Such tools could be derived from neurophysiological measurements of disease.

**Methods and analysis** The New Therapeutics in Alzheimer's Disease study (NTAD) aims to identify a biomarker set from magneto/electroencephalography that is sensitive to disease and progression over 1 year. The study will recruit 100 people with amyloid-positive mild cognitive impairment or early-stage Alzheimer's disease and 30 healthy controls aged between 50 and 85 years. Measurements of the clinical, cognitive and imaging data (magnetoencephalography, electroencephalography and MRI) of all participants will be taken at baseline. These measurements will be repeated after approximately 1 year on participants with Alzheimer's disease or mild cognitive impairment, and clinical and cognitive assessment of these participants will be repeated again after approximately 2 years. To assess reliability of magneto/electroencephalographic changes, a subset of 30 participants with mild cognitive impairment or early-stage Alzheimer's disease will also undergo repeat magneto/electroencephalography 2 weeks after baseline. Baseline and longitudinal changes in neurophysiology are the primary analyses of interest. Additional outputs will include atrophy and cognitive change and estimated numbers needed to treat each arm of simulated clinical trials of a future disease-modifying therapy.

**Ethics and data statement** The study has received a favourable opinion from the East of England Cambridge Central Research Ethics Committee (REC reference 18/EE/0042). Results will be disseminated through internal reports, peer-reviewed scientific journals, conference presentations, website publication, submission to regulatory authorities and other publications. Data will be made available via the Dementias Platform UK Data Portal on completion of initial analyses by the NTAD study group.

## INTRODUCTION

With 44 million people living worldwide with dementia, treatments to slow or stop disease

## STRENGTHS AND LIMITATIONS OF THIS STUDY

⇒ New Therapeutics in Alzheimer's Disease (NTAD) is a longitudinal, multicentre study of magneto/electroencephalographic measures of Alzheimer's disease progression.
⇒ NTAD assesses the test–retest reliability of magneto/electroencephalographic parameters.
⇒ All participants with early-stage Alzheimer's disease or mild cognitive impairment will be amyloid positive.
⇒ Attrition during follow-up may limit inferences.
⇒ Recruitment is from volunteer panels and clinical services that may not reflect national diversity.

progression are urgently needed. Alzheimer's disease is the most prevalent dementia, but despite rapid advances in therapeutics within preclinical models,[1][2] clinical trials remain expensive, slow and challenging[3–5] with high-profile failures.[6] Significant bottlenecks exist in early-phase trials, where efficacy rates are low and costs rapidly escalate. To bridge the gap between animal models and the human disease, better tools are needed to quantify pathogenic and pathophysiological mechanisms in patients in vivo providing evidence to pursue or discontinue trials of a candidate treatment. These tools should be safe, scalable and able to support early-phase trials over a short duration and viable budget.

Brain imaging is widely used to diagnose dementia and measure pathology in clinical trials. There are diverse imaging methods with differential sensitivity to brain structure, chemistry, pathology and function. For example, MRI is commonly used to measure structural changes related to Alzheimer's disease,[7] such as entorhinal cortex[8] and hippocampus volumes.[9] Positron emission tomography (PET) can quantify and localise Alzheimer pathology through the use of

BMJ

ligands that bind to the aggregated τ protein,[10–14] beta-amyloid plaques,[15] synapses[16] and neuroinflammation.[17]

Loss of synapses and synaptic plasticity is an early feature of Alzheimer's disease.[18–20] Indeed, cognitive decline may be more directly related to loss of synapses or synaptic plasticity than to atrophy.[21 22] As synaptic changes occur early in Alzheimer's disease,[21] synaptic measures could be sensitive to the earliest precursors of cognitive involvement. This accords with preclinical models in transgenic mice, where network physiology and cognition are impaired before tangles or cell death.[23]

In contrast to preclinical models and postmortem analysis, there are limited options to assess synaptic function in humans, in vivo. While PET imaging now offers ligands that indirectly quantify synaptic density, electroencephalography (EEG) and magnetoencephalography (MEG) measure neurophysiological properties dependent on synaptic integrity and function. MEG, for example, can identify synaptic and local circuit impairments[24 25] and their impact on network dynamics in Alzheimer's disease,[26–29] frontotemporal dementia[25 30–33] and Lewy-body disease.[34] MEG and EEG (M/EEG), therefore, have potential to support and derisk clinical trials of novel compounds. To enable M/EEG as viable biomarkers in clinical trials, one must assess their test–retest reliability.

The New Therapeutics in Alzheimer's Disease (NTAD) study aims to identify viable MEG and EEG biomarkers for clinical trials. NTAD is a multicentre study established by the Dementias Platform UK and supported by the Medical Research Council, Alzheimer's Research UK and industry partners.

This paper describes the NTAD protocol to acquire longitudinal MEG and EEG data in people with biomarker-positive mild cognitive impairment or early Alzheimer's disease.

## Research aims

The primary objective is to identify a biomarker set from neurophysiological M/EEG sensitive to the presence and progression of Alzheimer's disease, in support of experimental medicine studies. We aim to harmonise M/EEG protocols across sites to allow biomarker identification from pooled data. Such a neurophysiological biomarker, or biomarker set should be related to cognitive function, have high test-retest reliability and track disease progression over the duration of clinical trials. Sensitivity to disease progression should ideally outperform current widely used biomarkers such as MRI and cognitive tests. Our secondary objective is to identify a biomarker set that can predict disease progression, explain individual differences in the future disease trajectory and variation in cognitive decline over time. This paper describes the study design, including participants and methods for data acquisition, harmonisation and sharing.

## Analysis

Investigators across sites will identify harmonised M/EEG preprocessing pipelines to reduce analytical variance.

Following standardised preprocessing, investigators will address the study aims with multiple analytical approaches, which will be detailed in future, peer-reviewed papers. Where appropriate, investigators are encouraged to preregister analyses. Analyses will follow an overarching framework:

Stage 0 (descriptive) analyses will describe demographics; Apolipoprotein E genotype; blood-based, fluidic biomarkers; neurocognitive profiles and regional brain atrophy. This serves to confirm the expected patterns of cognitive decline and atrophy typical of people with mild cognitive impairment and early Alzheimer's disease and to enable comparison of the NTAD cohort across sites and with other study populations.

Stage 1 (baseline) analyses will identify reliable and sensitive baseline, disease-related M/EEG parameters in terms of (1) the sensitivity of cross-sectional M/EEG parameters to group differences, (2) the correlation of cross-sectional M/EEG parameters to baseline cognition, atrophy and fluidic markers and (3) the test–retest reliability of M/EEG parameters.

Stage 2 (longitudinal) analyses will assess the sensitivity of selected neurophysiological markers to longitudinal change by assessing how they change over ~12 months. Sensitivity (as accuracy and effect sizes) to disease progression can be compared across modalities.

Stage 3 (predictive) analyses aim to identify baseline parameters that predict disease progression and identify the optimal parameter, or parameter set, that provides parsimonious or unique predictive information.

Potential parameters at stages 1–3 include: (1) summary statistics of evoked and induced responses for each task, (2) summary statistics of the spectral density in resting-state recordings, (3) data-driven connectivity metrics, including those derived from hidden Markov modelling[35–37] and (4) parameters of model-based analyses of evoked responses and spectral densities using dynamic causal modelling.[25 38 39] The data set will be suitable for mediation modelling to determine how longitudinal changes in brain structure and neurophysiology explain cognitive changes. Full details of these analyses will be described in their respective preregistration reports and peer-reviewed publications. With data sharing, new analytical methods may be applied to this data set alone or in combination with other cohorts' data, for example, ENIGMA-MEG (http://enigma.ini.usc.edu/ongoing/enigma-meg-working-group/) and BIOFIND (https://biofind-archive.mrc-cbu.cam.ac.uk/).

DNA will allow the description of genotype, including common genetic variants, and post hoc identification of rare, autosomal dominant cases which may have distinctive physiology and phenotype. Potential analyses could examine the relationship between neurophysiological changes and (1) Alzheimer's disease risk alleles (eg, apolipoprotein e4), (2) polygenic risk scores of Alzheimer's disease and (3) plasma τ levels; noting that the study is not designed for adequate power for genome-wide associations except as a contribution to pooled analyses. RNA is collected to enable post hoc transcriptomics analysis.

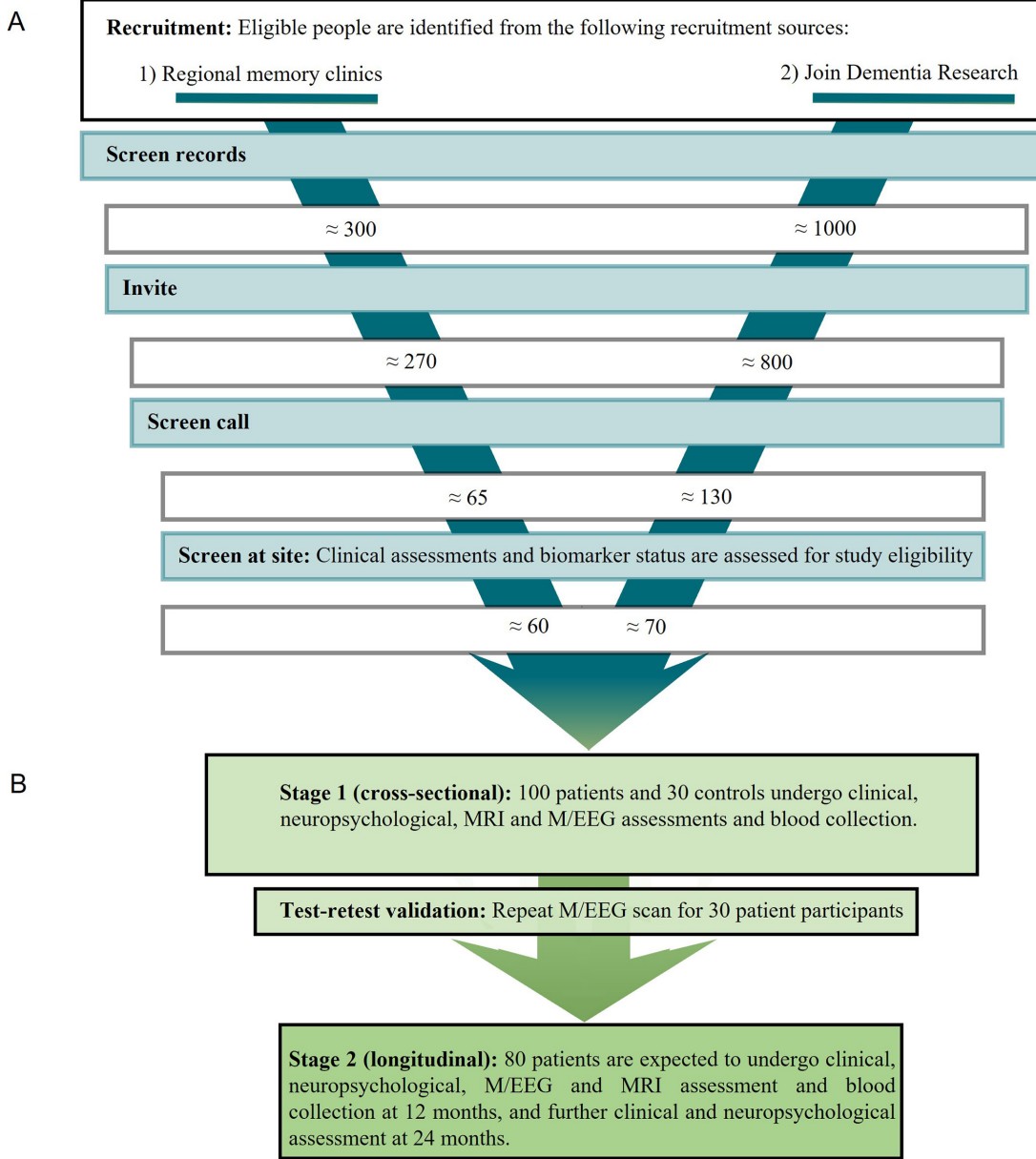

A

Recruitment: Eligible people are identified from the following recruitment sources:

1) Regional memory clinics                                        2) Join Dementia Research

**Screen records**

≈ 300                                                            ≈ 1000

**Invite**

≈ 270                                                            ≈ 800

**Screen call**

≈ 65                                                            ≈ 130

**Screen at site:** Clinical assessments and biomarker status are assessed for study eligibility

≈ 60         ≈ 70

B

**Stage 1 (cross-sectional):** 100 patients and 30 controls undergo clinical, neuropsychological, MRI and M/EEG assessments and blood collection.

**Test-retest validation:** Repeat M/EEG scan for 30 patient participants

**Stage 2 (longitudinal):** 80 patients are expected to undergo clinical, neuropsychological, M/EEG and MRI assessment and blood collection at 12 months, and further clinical and neuropsychological assessment at 24 months.

**Figure 1** Participant flow chart. (A) Recruitment strategy: electronic screening of records from regional memory clinics and Join Dementia Research generates approximately 300 potential patient and 1000 potential patient and control participants who are invited to telephone screening. People who remain eligible are screened further on-site to identify 100 patient and 30 control participants. (B) Study stages: stage 1 consists of the cross-sectional baseline assessments and, for 30 patients, a repeat of the M/EEG assessment 2 weeks after the first. Only patients continue to the longitudinal stage 2 of the study, with repeat assessments 12 and 24 months after baseline. Attrition of 20% is expected at stage 2. M/EEG, Magnetoencephalography combined with electroencephalography imaging.

## METHODS
### Study design
This is a repeated-measures, observational-design study with 130 participants tested at two sites: Cambridge University and Oxford University, each in partnership with local NHS Hospital Trusts. The planned start and end dates are 1 June 2018 and 1 July 2023. The protocol includes two stages (figure 1). The first, cross-sectional stage consists of baseline and test–retest sessions. Baseline sessions are comprised of clinical and neuropsychological assessments, an MRI scan and an M/EEG scan. For the test–retest session, a subset of patients returns for a second M/EEG approximately 2 weeks after the first. Patients proceed to stage 2, with repeat assessment at 12 months (including M/EEG, MRI, clinical and neuropsychology reassessments) and 24 months (clinical and neuropsychological reassessments).

### Participant recruitment and selection
Participants are aged between 50 and 85 years with similar numbers of men and women, with symptomatic mild cognitive impairment or early Alzheimer's disease (n=100) or normal cognition (n=30, figure 1). Potential participants are identified by a staged

screening process. First, approved clinical staff undertake electronic screening of records held by regional memory clinics or NIHR Join Dementia Research, or research team members review electronic records of people who previously consented to data retention on a contact list regarding dementia research opportunities. A participant information sheet detailing the study procedures is provided to candidate participants and a study partner who regularly sees the participant and is willing to complete assessments. Candidate participants answer a series of questions derived from the inclusion and exclusion criteria (table 1) during a telephone-based screening call.

Potentially eligible participants are invited to onsite screening (see below). For those with Alzheimer's disease or mild cognitive impairment, clinical letters are sought from their memory assessment service, in addition to within-study assessments. This final screening appointment allows further assessment of eligibility, against exclusion and inclusion criteria (table 1) and Alzheimer biomarker status. Any uncertainty regarding eligibility is discussed with the principal investigator for the relevant site. Eligible people proceed to stage 1.

### Sample size and power

Previously published studies have detected the presence and progression of Alzheimer's disease or mild cognitive impairment with under 100 participants using MRI.[40] We aim to identify MEG measures that are at least as sensitive as MRI. For longitudinal analysis, MEG[41–43] and EEG[44–49] have been used over 11–36 months with 20–130 participants. Mandal and colleagues[50] summarise the accuracies of several MEG measures to distinguish Alzheimer's disease from controls: 30/34 MEG measures had areas under the curve >0.64 approximating (under Gaussian assumptions) to Cohen's d >0.5.[51]

Power analyses were conducted in G*power V.3.1,[52] to determine effect sizes detectable with 80% power and 5% error rate. For the cross-sectional NTAD data with a group of 100 patients and 30 controls, a one-tailed, independent t test can detect medium effect sizes d>0.52, while one-tailed correlations with disease severity in the patient group can detect effect sizes of R>0.24. Assuming 20% attrition at follow-up (n=80), longitudinal analyses of disease progression in the patient group would have 80% power to detect within-sample differences of d>0.28 (figure 2). Future preregistration and analysis reports will undertake specific power estimations.

### Patient and public involvement

The use and tolerability of M/EEG in combination with the PET or cerebrospinal fluid, MRI and neuropsychology were discussed as part of Alzheimer Society-sponsored patient groups with patients and carers affected by AD or MCI for the closely related

**Table 1** Group inclusion criteria and general exclusion criteria

| Group inclusion criteria | |
|---|---|
| **Patients** | **Controls** |
| Diagnosis of MCI or AD* | No neurological diagnosis |
| CDR=0.5–1 | CDR=0 |
| Positive AD biomarker status (CSF or PET) | Known AD biomarker status (CSF or PET) |
| MMSE >18 | MMSE>24 |
| 50–85 years | 50–85 years |
| **General exclusion criteria (patients and controls)** | |
| Significant neurological disease, other than Alzheimer's disease, that may affect cognition or ability to complete the study | |
| Presence of any significant psychiatric disorder that could affect participation[85] | |
| Any clinically important abnormality that could compromise study participation | |
| A clinically significant illness, medical or surgical procedure, or trauma within 30 days prior to screening or baseline | |
| Known or suspected systemic infection | |
| Medications affecting cognition, unless on a stable dose for>30 days prior to baseline | |
| Rosen Modified Hachinski Ischaemic score≥4 | |
| History of seizure, except febrile seizures or single provoked seizure | |
| Head trauma resulting in protracted loss of consciousness, or serious infectious disease affecting the brain, within 5 years of screening and baseline Participation in a clinical trial of an investigational medicinal product | |
| Impairment of vision or hearing that could affect study participation | |
| Formal education ≤7 years | |
| Lack of mental capacity or other ability to consent | |
| Inability to read and write fluently in English | |
| Inability to walk 10 metres independently† | |
| Contraindications to blood sampling | |
| Contraindications to lumbar puncture (eg, spinal deformations) and amyloid PET scan | |
| Contraindication to MRI (including, but not limited to, claustrophobia; pregnancy; MR-incompatible pacemakers and other MR-incompatible, implanted medical devices) | |
| Metallic implants in the body that affect MEG recordings, as judged by the Investigator | |

*According to Albert et al[86] or McKhann et al.[55]
†Ten metre walking is required for practicality and safety during the scanning sessions to avoid the use of walking aids, which may have metallic constituents.
AD, Alzheimer's disease; CDR, clinical dementia rating; CSF, cerebrospinal fluid; MCI, mild cognitive impairment; MMSE, Mini Mental State Examination; PET, positron emission tomography.

Dementias Platform UK study known as 'Deep and Frequent Phenotyping' with which NTAD is aligned in design and methods. In addition, the use of MEG for dementia research featured in the public engagement event 'MEG and me' as part of the Cambridge Science Festival and has been discussed at open meetings at the Cambridge Science Festival and Alzheimer

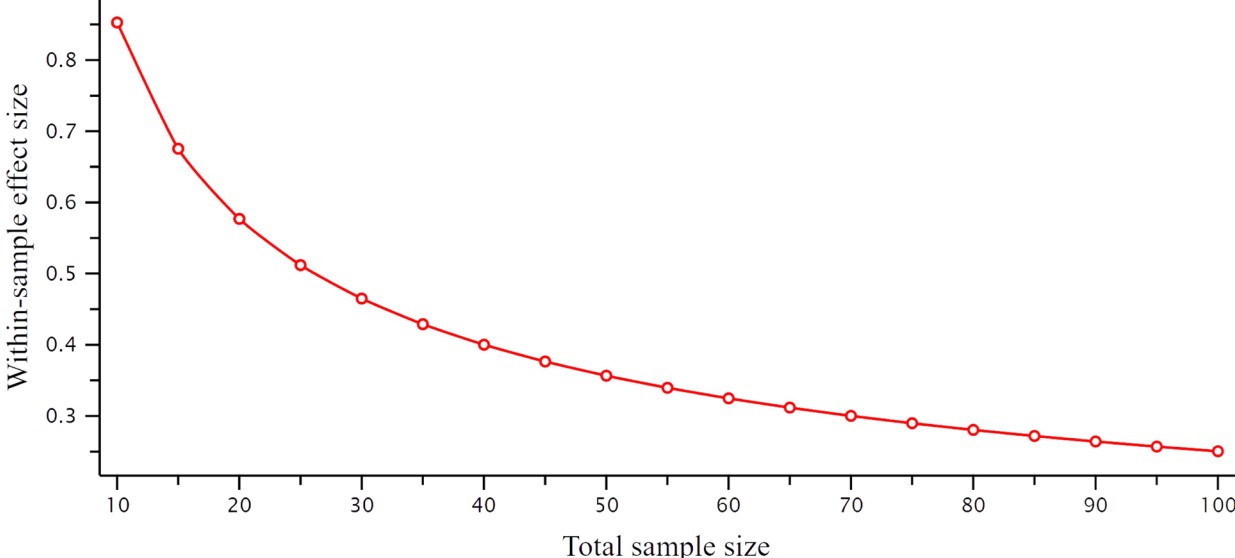

**Figure 2** Effect sizes required to detect longitudinal correlations with disease severity at different attrition rates (resulting in reduced longitudinal sample size) with 80% power and α=0.05. Effect sizes were estimated using G*power V.3.1 software for one-tailed, paired t-tests. We anticipate <20% attrition per annum.

Research UK Eastern Region network public meetings. The study is also cofunded by Alzheimer Research UK, a charity with a joint patient advocacy and research funding remit.

## STUDY PROCEDURES
### Overview of protocol stages
#### Screening
Potential participants are identified from electronic records and screened further via telephone call. Prospective participants who remain eligible are invited to an in-person screening appointment. Any further questions they have are answered before they provide written, informed consent. After written, informed consent, a medically qualified clinician administers the clinical interview, clinical dementia rating (CDR) and Haschinski ischaemic scale. In the context of this study, medically qualified means having at least the Membership of the Royal Colleges of Physicians of the United Kingdom part II qualifications and in ST3+ level neurology or psychiatry training or completion of neurology or psychiatry specialist training. The remaining clinical assessments are completed by a member of the research team and include physiology and blood sampling. All research team members have current Good Clinical Practice training. If Alzheimer biomarker status is unknown from recent clinical or research assessments, participants proceed to either cerebrospinal fluid examination or amyloid PET imaging according to participant preference and eligibility. For patients, where a participant's positive amyloid status has been confirmed previously, a positive result enables participation, a negative result excludes participation. For controls, the Alzheimer biomarker status must be known to proceed but, given their CDR=0, a positive test does not prevent participation.

#### Stage 1: baseline
One hundred people with mild cognitive impairment or Alzheimer's disease and 30 neurologically normal people proceed to baseline assessment. Participants undergo structured neuropsychological assessment, M/EEG and MRI over two sessions (or three by preference).

#### Stage 1: 2-week M/EEG retest
We invite 30 people from the patient group (ie, mild cognitive impairment or Alzheimer's disease) to repeat the M/EEG scan between 2 and 4 weeks after the first scan. Invitations are prioritised to people who can most readily attend the additional session (eg, considering distance) until the target sample size is reached.

#### Stage 2: annual follow-up 1
Participants in the patient group repeat the clinical and neuropsychological assessments, M/EEG scan, MRI scan and blood collection at 12 months after baseline.

#### Stage 2: annual follow-up 2
Clinical and neuropsychological assessments are repeated at 24 months for participants in the patient group.
The study timeline is illustrated in figure 3.

### Blood samples
Participants are asked to consume only water for 2 hours prior to blood collection. At baseline and follow-up, blood is drawn in the following order: 2.7 mL in sodium citrate tubes (plasma), 5 mL in serum separator tubes (serum), 10 mL in EDTA tubes (DNA), 10 mL in EDTA tubes (plasma and buffy coat) and 2×2.5 mL in PAXgene tubes (RNA). Once filled, the EDTA and PAXgene tubes are inverted 10 times. From April 2021, 4.9 mL of blood is collected in S-monovette tubes for SARS-CoV-2 serology.

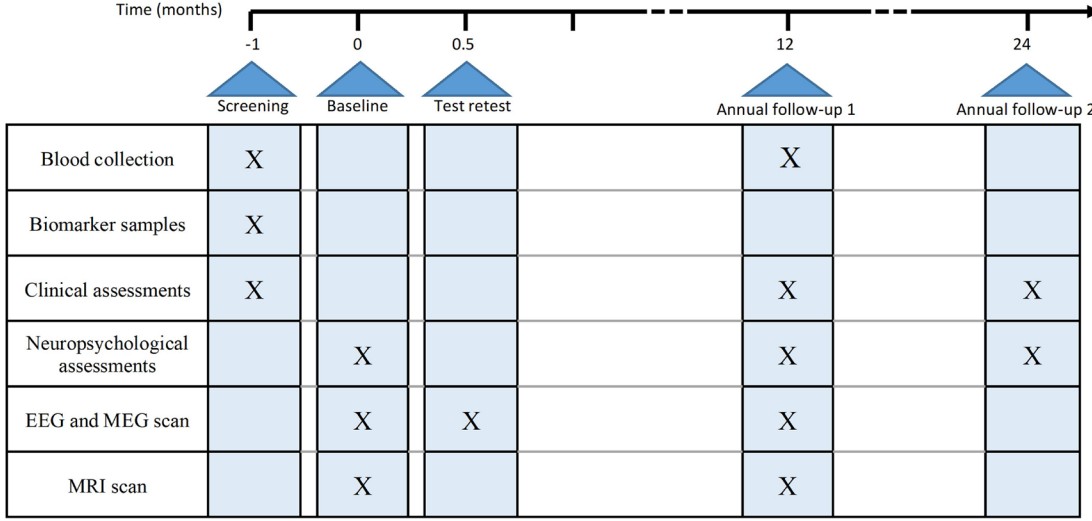

**Figure 3** Timeline summary of study procedures. EEG, electroencephalography; MEG, magnetoencephalography.

Samples are processed and frozen according to the guidelines in table 2.

## Biomarkers for Alzheimer disease pathology

Clinical criteria are insufficient to reliably diagnose the presence of Alzheimer's disease pathology.[53][54] We seek additional biomarker evidence, using either cerebrospinal fluid or PET imaging according to participant preference and eligibility.[55]

### Cerebrospinal fluid

Cerebrospinal fluid is obtained by lumbar puncture and collected in polypropylene tubes. Within 1 hour of collection, cerebrospinal fluid is centrifuged, separated and the supernatant frozen to −80°C for later batch analysis using a chemiluminescent enzyme immunoassay for total τ, phosphorylated τ and amyloid beta 1–42 levels. Positive amyloid status is indicated by a total τ to amyloid beta 1–42 ratio >1, and amyloid beta 1–42 concentration <450 pg/mL. pTau181 concentration is recorded from April 2021.

### Amyloid PET

Participants receive a 300 MBq bolus injection of florbetaben and are scanned 80–100 min postinjection on a GE Signa PET/MR scanner at the Wolfson Brain Imaging Centre, Cambridge or a GE D710 PET/CT scanner at the Churchill Hospital, Oxford. The centiloid method is used to classify the florbetaben scans as amyloid positive with centiloid >1.19.[56][57]

## CLINICAL ASSESSMENTS

Clinical assessments are completed at a clinical research centre in one or two sessions.

### Clinical interview

A study clinician interviews the participant and study partner. Clinicians follow a structured interview that covers sociodemographic factors, including age and years of education; lifestyle factors; family history of dementia; medical history, including information on significant medical conditions with date of onset; and concomitant medication usage, with details about dosage and duration.

### Addenbrooke's cognitive examination

The revised Addenbrooke's cognitive examination evaluates orientation, memory, verbal fluency, language and visuospatial domains.[58] It is administered and scored according to the Administration and Scoring Guide (2006). Alternate versions at each visit reduce practice effects.

### Mini-Mental State Examination

The Mini-Mental State Examination is acquired.[59]

### Clinical dementia rating

The Clinical Dementia Rating quantifies dementia severity through a structured interview.[60–62] Interviewers

| Table 2 | Blood sample processing guidelines |
|---|---|
| Plasma | The sample in the sodium citrate tube is centrifuged at 2000 g for 10 min at room temperature. The plasma is aliquoted into 1.4 mL matrix tubes and stored at −80°C. |
| Serum | At least 30 min after blood collection, the sample in the serum separator tube is centrifuged at 2000 g for 10 min at room temperature. The serum is aliquoted into 1.4 mL matrix tubes and stored at −80°C. Similarly, the sample in the S-monovette tube is centrifuged at 2000 g for 10 min at room temperature at least 30 min after collection. The sample is aliquoted into 2 mL Sarstedt tubes. |
| DNA | The first EDTA tube is frozen as soon as possible and stored at −80°C for later DNA extraction and analysis. |
| Plasma and buffy coat | The second EDTA tube is centrifuged immediately at 2500 g for 15 min at room temperature. The plasma and buffy coat are aliquoted into separate 1.4 mL matrix tubes and stored at −80°C. |
| RNA | Paxgene tubes are stored at room temperature for 2 hours before being frozen upright at −20°C for 24 hours and then stored at −80°C. |

rate participant impairments in six categories: memory, orientation, judgement and problem solving, community affairs, home and hobbies and personal care. The global CDR is derived from these ratings.[61]

## Haschinski ischaemic score

The Rosen modification of Haschinski's ischaemic score seeks to differentiate between primary progressive dementias, including Alzheimer's disease and multi-infarct dementia.[63] A study clinician uses information from medical history, physical and neurological examination and medical records to determine the score. Scores below 4 indicate a low likelihood of vascular disease as the cause of dementia.

## Self-reported questionnaires

Participants complete the 30-item Geriatric Depression Scale,[64] 40-item Spielberger State-Trait Anxiety Inventory[65] and 11-item Pittsburgh Sleep Quality Index.[66] Study partners complete the 30-item Amsterdam Instrumental Activity of Daily Living Questionnaire (short version)[67] and Mild Behavioral Impairment Checklist.[68]

## Physiological measures

Height, weight, hip-waist ratio, blood pressure, pulse rate and temperature are measured for participants using a stadiometer, electronic weight scales and stretch-resistant tape.

## Neuropsychological assessment

The neuropsychological test battery closely resembles the Deep and Frequent Phenotyping study[69] and IMI-European Prevention of Alzheimer's Disease,[70 71] including the Pre-Alzheimer Cognitive Composite.[72] Assessments take place in a private, testing room in one session with breaks as needed, but two breaks by default (see table 3).

## Repeatable battery for the assessment of neuropsychological status

Repeatable battery for the assessment of neuropsychological status (RBANS) measures cognitive decline in five domains: immediate memory, visuospatial, language, attention and delayed memory.[73] Participants receive alternate forms at repeated assessments. The immediate memory index comprises the *list learning subtest*, with immediate recall of 10-items over four trials and the *story memory subtest*, with immediate recall of a 12-item story over two trials. The visuospatial index comprises the *figure copy subtest*, which involves copying a geometric figure and the 10-item, *line orientation subtest*. The language index comprises the 10-item, *picture naming subtest* and the *semantic fluency subtest*, where participants name as many exemplars of the given semantic category as possible in 60 s. The attention index comprises the *digit span forwards subtest*, involving immediate repetition of increasing digit strings and the *coding subtest*, scored as the total number of correctly coded numbers generated using an item-to-number code within 90 s. The delayed memory index comprises the *list recall subtest*, involving free recall of the

**Table 3** Order of neuropsychological assessments

| Order of assessments | Duration (min) |
| --- | --- |
| 1. List learning (RBANS subtest) | ≈ 40 |
| 2. Story memory (RBANS subtest) | |
| 3. Figure copy (RBANS subtest) | |
| 4. Line orientation (RBANS subtest) | |
| 5. Picture naming (RBANS subtest) | |
| 6. Semantic fluency (RBANS subtest) | |
| 7. Digit span forwards (RBANS subtest) | |
| 8. Digit span backwards | |
| 9. Coding (RBANS subtest) | |
| 10. List recall (RBANS subtest) | |
| 11. List recognition (RBANS subtest) | |
| 12. Story recall (RBANS subtest) | |
| 13. Figure recall (RBANS subtest) | |
| *Break* | ≈ 15 |
| 14. Free and cued selective reminding test (PACC) | ≈ 45 |
| 15. Logical memory part 1 (PACC) | |
| 16. National adult reading test | |
| 17. Digit symbol substitution (PACC) | |
| 18. Trails B | |
| 19. Logical memory part 2 (PACC) | |
| *Break* | ≈ 15 |
| 20. Reaction time task (CANTAB subtest) | ≈ 30 |
| 21. Paired associates learning (CANTAB subtest) | |
| 22. Rapid visual processing (CANTAB subtest) | |
| 23. Spatial working memory (CANTAB subtest) | |
| 24. Four mountains task | |

CANTAB, Cambridge Neuropsychological Test Automated Battery; PACC, Pre-Alzheimer Cognitive Composite; RBANS, repeatable battery for the assessment of neuropsychological status.

list learning task; *list recognition*, where participants decide whether a word was included in the list learning task; *story recall*, where participants freely recall the story memory task; and *figure recall*, where participants draw from memory the figure presented in the figure copy task.

## Digit span backwards

Digit span backwards, from the Wechsler adult intelligent scale,[74] is used in conjunction with RBANS digit span forwards. It follows the RBANS digit span format, but participants are asked to immediately repeat in reverse order.

## Free and Cued Selective Reminding Test

The Free and Cued Selective Reminding Test assesses episodic memory and distinguishes retrieval from storage deficits.[75] Sixteen pictured items are encoded during the initial learning phase, where participants identify and name items responding to unique semantic cues. After a short delay, participants freely recall all items. Interviewers prompt for each item not recalled using the

unique semantic cues from the learning phase. Participants receive alternate forms at repeated assessments.

### Logical memory

Logical memory, taken from the Wechsler memory scale (third edition), assesses episodic memory.[76] Participants immediately recall short stories they have been read. After a 30 min delay with intervening tests, participants freely recall the stories and answer yes or no questions testing story recognition.

### National Adult Reading Test

The national adult reading test (second edition) estimates premorbid intelligence from printed, irregular words.[77]

### Digit symbol substitution

The digit symbol substitution from the Wechsler Adult Intelligence Scale assesses processing speed and attention.[74] Participants have 90 s to code as many correct symbols as possible corresponding to presented numbers by using the given number-to-symbol code.

### Trails Making Test B

The Trails Making Test B assesses executive function, attention and processing speed.[78] Following a practice sample to ensure task comprehension, participants are presented with 25 encircled numbers on a page, which they connect by alternating between numbers and letters. The time it takes to complete the sample and any errors made are recorded.

### Cambridge Neuropsychological Test Automated Battery

The tablet-based Cambridge Neuropsychological Test Automated Battery assesses processing speed, episodic memory, attention, working memory and executive function.[79] The reaction time task (simple and five choice variant) assesses processing speed. Participants hold down a response button and release this to respond to on-screen targets. The paired associates learning task (standard variant) assesses episodic memory. Participants learn associations between patterns and their locations. An initial learning stage precedes immediate recall. The rapid visual processing task (three-target variant) assesses attention. Single digits appear on the screen. Participants respond when they see a string matching target sequences. The spatial working memory task (standard variant) assesses working memory and executive function. Participants search inside on-screen boxes to find and collect tokens. Tokens never appear in the same box two times.

### Four mountains task

The tablet-based four mountains task assesses allocentric spatial processing.[80] During a learning phase, participants learn the topographical layout of four mountains presented in a computer-generated landscape. Following a delay, participants are presented with four alternative images and identify the target image matching the topographical mountain layout of the learning-phase image, but with potentially altered colours, textures and points of view.

### Neurophysiology (M/EEG)

M/EEG data are collected simultaneously at 1000 Hz in a magnetically shielded room. At Cambridge, data are collected using the Elekta VectorView system from 2017 to December 2019, with 204 planar gradiometers, 102 magnetometers and a 70-channel Easycap EEG. Stage 1 scans and the first 12 follow-up scans used the same scanner. The MEGIN Triux Neo M/EEG scanner is used from March 2020 onwards with the same sensor configuration as the VectorView and a 64-channel Easycap. At Oxford, the MEGIN Triux Neo M/EEG scanner and an EasyCap 60 channel BrainCap for MEG with an augmented 10/20 layout are used for all data collection.

The Polhemus digitisation system records the position of the standard fiducial points, >300 additional head points, five head position indicator coils and EEG electrodes. Head position indicator coils measure head position within the MEG helmet. Electrodes on the right clavicle and left lower rib record ECG data. Electrodes above and below the left eye and on bilateral canthi record vertical and horizontal electro-oculogram data. Reference and ground electrodes are placed on the left side of the nose and left cheek, respectively. During the seated scan, participants rest or perform simple tasks, responding through a button box. Participants wear non-magnetic earphones with sound delivered though plastic tubes and, if necessary, non-magnetic glasses. Prior to M/EEG, a Snellen eye test and pure tone audiometry assess sight and hearing thresholds. Tasks are performed in the following order.

### Simple audio-visual task

Participants fixate on a red, central fixation dot and quickly respond to each auditory or visual stimuli with a button press (figure 4A). Auditory tones (n=100) of 300 Hz, 600 Hz or 1200 Hz are presented for 300 ms after a blank interval of 1000 ms. Visual stimuli are concentric black and white circles that appear for 300 ms after 3000 ms. Filler trials (n=30) containing only the red fixation dot are included. Visual and auditory trials are randomly intermixed. Ten initial practice trials familiarise participants to the task.

### Auditory mismatch-negativity task

The roving auditory mismatch negativity task elicits error responses to deviant tones followed by rapid plasticity as predictions are updated on repetition of the deviant stimulus.[81 82] Participants passively watch a muted nature documentary. Through earpieces, they hear binaural, in-phase sinusoidal tones >60 dB above the average auditory threshold with durations of 100 ms and 500 ms stimulus onset asynchrony. Tone frequencies are the same within but different between blocks. Blocks range from 400 Hz to 800 Hz. The number of tones per blocks vary from 3 to 11 according to a truncated exponential distribution (figure 4B).

**Figure 4** (A) Audio-visual task: participants visually fixate on a red dot and press a button whenever they hear or see something. (B) Mismatch-negativity task: participants passively watch a nature documentary while listening to tones through the earpieces. Red dashes represent deviant tones and black dashes represent standard tones (from the sixth repetition). (C) Scene repetition task: participants view scenes and press the button when they see a scene containing a moon. Each non-target scene repeats once during the task. Images in the figure are for illustrative purposes only. (D) cross-modal oddball task: participants view object-tone pairs consisting of four learnt (standard) pairs (the standard sounds are black in the figure); associative-deviant, object-tone pairs (the associative deviant sound is coloured red in the figure); and standard objects presented with novel tones (the novel deviant tones are blue in the figure). Participants press a button when they see the letter 'a'. The visual objects were designed by Freepik.

### Scene-repetition task

In this passive memory test, participants view a series of complex scenes (landscapes and cityscapes) and press a button only when scenes contain a moon (n=26, different moon images). Target scenes containing a moon ensure attention but are not of interest; the main interest is difference between initial and repeated presentations of non-target scenes. Each scene is presented for 800 ms and preceded by a fixation cross of 200 ms on average (100–300 ms). Scenes are pseudo-randomly intermixed with the constraint that 10 initial 'burn-in' scenes, with two moon target scenes, precede 256 scenes presented two times, with 14–93 (median=42) intervening scenes between the first and repeat presentation (figure 4C).

### Cross-modal Oddball task

The task assesses hippocampal-dependent, paired-associates learning.[83 84] Trials comprise of a 700 ms visual, abstract 'object' and a 400 ms sound, starting 300 ms after trial onset. Interstimulus intervals average 300 ms. During the initial

**Table 4** Order and parameters of MRI sequences

| | |
|---|---|
| T1-weighted | The 3D T1-weighted structural image is acquired using the generalised autocalibrating partially parallel acquisitions (GRAPPA) technique applied to a high-resolution magnetisation prepared rapid gradient echo (MPRAGE) sequence with the following parameters: TE=2.91 ms, TR=2300.00 ms, TI=900.00 ms, flip angle=9 degrees, acquisition matrix=256×240, voxel size=1.00 mm isotropic, number of slices=176, slice thickness=1 mm, acquisition time=5 min and 12 s |
| T2 FLAIR | The 3D FLAIR structural image is acquired using the following parameters: TE=394.00 ms, TR=5000.00 ms, TI=1650.00 ms, flip angle=120 degrees, acquisition matrix=256×256, voxel size=1.00 mm isotropic, number of slices=192, slice thickness=1.00 mm, acquisition time=5 min and 7 s |
| T2*-weighted | The T2*-weighted structural image is acquired using the following parameters: TE=20.00 ms, TR=640.00 ms, Flip angle=20 degrees, acquisition matrix=256×256, number of slices=47, slice thickness=3.00 mm, Voxel size=0.90×0.90×3.00mm, Acquisition time=4 min and 7 s |
| T2-weighted with fat saturation | The T2-weighted image with fat saturation is acquired with the following parameters: TE=78.00 ms, TR=4000.00 ms, flip angle=150 degrees, acquisition matrix=256×232, number of slices=47, slice thickness=3.00 mm, voxel size=0.90×0.90 x3.00mm, acquisition time=3 min and 26 s |
| Diffusion-weighted imaging | The diffusion-weighted images are acquired with a spin-echo echo-planar sequence. B-values of 0, 300, 700 and 2000 s/mm$^2$ were used with 116 diffusion gradient directions obtained via MRTrix3.[87] The number of signal averages was 12, 8, 32 and 64 for b-values of 0, 300, 700 and 2000, respectively. The acquisition is run in the transverse orientation without any angulation. Other scan parameters include: TR=3800 ms; TE=85.00 ms, voxel size=2.50 mm isotropic, acquisition matrix=96×96, number of slices=64, slice thickness=2.50 mm, acquisition time=8 min and 45 s |
| Quantitative susceptibility mapping | The quantitative susceptibility mapping correlates with Aβ-amyloid iron density and is acquired with a 3D gradient echo sequence with multiple echoes and flow compensation in the readout and slice directions. (TE 1=5.20 ms, TE 2=10.4 ms, TE 3=15.6 ms, TE 4=20.8 ms, TE 5=26.0 ms; TR=31.0 ms, flip angle=15 degrees, acquisition matrix=256×192, number of slices=160 (per TE), slice thickness=1.00 mm, voxel size=1.00 mm isotropic, acquisition time=5 min 10 s. |
| Resting state eyes open | For the resting-state scan, the participants are presented with a fixation cross and asked to maintain fixation throughout the task. Echo planar images are acquired with 200 volumes with the following parameters: TE=30.00 ms, TR=1500.00 ms, flip angle=90 degrees, acquisition matrix=64×64, number of slices=54, slice thickness=3.00 mm, voxel size=3.00 mm isotropic, acquisition time=5 min 50 s |
| Hippocampal subfields | The high-resolution, hippocampal subfield images are acquired with T2-weighted, turbo spin echo sequences and run with the following parameters: TE=50.00 ms, TR=8020.00 ms, flip angle=122 degrees, acquisition matrix=448×448, number of slices=30, slice thickness=2.00 mm, voxel size=0.4×0.4 x2.0mm, acquisition time=8 min 11 s. |
| Arterial spin labelling | 3D arterial spin labelling images are acquired using a flow-sensitive alternating inversion recovery spin-echo pulsed sequence with Q2-TIPS bolus saturation in the transverse plane with the following parameters: TR=4000.00 ms; TE=13.20 ms; flip angle=130 degrees, labelling duration=800 ms, post-labelling delay=2000 ms, acquisition matrix=64×60; slices per slab=32, slice thickness=4.50 mm, voxel size=1.85×1.85 x4.50mm, acquisition time=6 min 46 s. |

FLAIR, fluid-attenuated inversion recovery; TE, echo time; TI, inversion time; TR, repetition time.

training period (80 trials), participants learn associations between four standard pairs of visual objects and sounds. The main task has 770 bimodal trials and 40 unimodal trials, randomly intermixed. Bimodal trials consist of: standard, learnt object-sound pairs (n=670); standard objects paired with novel sounds (n=50) and mismatched pairs, where the object and sound are from different standard pairs (n=50). Unimodal trials ensure that the task is attended, participants press a button when they see the letter 'a' (figure 4D). A 10-item assessment follows the task, participants hear a standard sound and report which of four presented objects the sound was paired with most often during the task.

### Eye-open resting state
Participants are presented with a central fixation cross and instructed to clear their mind, relax, think of nothing specific and focus on the fixation cross for 5 min.

### Eye-closed resting state
Participants are instructed to clear their mind, relax, think of nothing specific and close their eyes but stay awake.

### MRI
MRI sequences use 3T Siemens PRISMA scanners at the Cambridge MRC Cognition and Brain Science Unit and Oxford Centre for Human Brain Activity (table 4).

## CURRENT STATUS
The study is active at both sites.

## ETHICS AND DISSEMINATION
The study received a favourable opinion from the East of England—Cambridge Central Research Ethics Committee (REC reference 18/EE/0042). Imaging data and clinical scores are hosted by Dementias Platform UK Imaging Platform (https://portal.dementiasplatform.uk), using XNAT (https://www.xnat.org). Data will be made available with a managed access process through Dementias Platform UK, subject to requesters agreeing to a Code of Conduct to preserve data security, confidentiality and privacy.

**Author affiliations**
[1]MRC Cognition and Brain Sciences Unit, University of Cambridge, Cambridge, UK
[2]Department of Clinical Neurosciences and Cambridge University Hospitals NHS Foundation Trust, Cambridge Biomedical Campus, University of Cambridge, Cambridge, UK
[3]Oxford Centre for Human Brain Activity, Wellcome Centre for Integrative Neuroimaging, Department of Psychiatry, University of Oxford, Oxford, UK
[4]Lilly Corporate Center, Indianapolis, Indiana, USA
[5]Lilly Centre for Clinical Pharmacology, Singapore
[6]Neuroscience, BioPharmaceuticals R&D, AstraZeneca, Cambridge, UK
[7]Department of Psychiatry, University of Oxford, Oxford, UK

[8]Cardiff University Brain Research Imaging Centre, School of Psychology, Cardiff University, Cardiff, UK

[9]Department of Psychiatry, University of Cambridge, Cambridge, UK

**Collaborators** Franklin Aigbirhio, Leeza Almedom, Richard Bevan-Jones, Lara Bolte, Haddy Fye, Negin Holland, Masud Husain, John Isaac, Amirhossein Jafarian, Hartmuth Kolb, Hilde Lavreysen, Alexander Murley, Edoardo Ostinelli, Holly Phillips, Ilan Rabiner, Alastair Reith, Alexander Shaw, Tony Thayanandan, Giacomo Salvadore, Katharine Smith, Duncan Street, David Whiteside and Caroline Zangani

**Contributors** JHL and JBR drafted the manuscript, with review and contribution from EK, AJQ, Y-JC, MK, JP, SL, MP, VR, KS, MW, ACN and RNH. JBR is the chief investigator of the study. RNH, ACN, MW, KDS and VR are principal investigators, contributing to conception and design. EK led protocol development and governance. JBR, MW, ACN, RNH, JI, SL, MP and KS conceived and designed the study.

**Funding** This work is part of the Dementias Platform UK (RG94383/RG89702, MR/T033371/1, MR/L023784/1 & MR/L023784/2) that is funded by the Medical Research Council, Janssen, AstraZeneca, Araclon, IXICO, Somalogic, GlaxoSmithKline, Invicro, Cambridge Cognition and Cognetivity. The study has additional support from Alzheimer's Research UK (ARUK-PG2017B-19), the Wellcome Trust (103838), Medical Research Council (SUAG/051 G101400, SUAG/046 G101400, SUAG/092 116768), NIHR Cambridge Biomedical Research Centre (BRC-1215–20014), NIHR Oxford Biomedical Research Centre (IS-BRC-1215–20008) and NIHR Oxford Health Biomedical Research Centre (IS-BRC-1215-20005). JHL is supported by the Medical Research Council Doctoral Training Programme. ACN is supported by the Wellcome Trust Senior Investigator Award (104571/Z/14/Z). The Wellcome Centre for Integrative Neuroimaging is supported by core funding from the Wellcome Trust (203139/Z/16/Z). The views expressed are those of the authors and not necessarily those of the NIHR or the Department of Health and Social Care. For the purpose of Open Access, the author has applied a CC BY public copyright licence to any Author Accepted Manuscript version arising from this submission.

**Competing interests** MP is employed by AstraZeneca and may currently hold AstraZeneca stocks or stock options. Y-JC and SL are employed by Eli Lilly and may currently hold Eli Lilly stock. MW receives royalties from FSL and is senior editor for Neuroimage. JBR is Chief Scientific Adviser for Alzheimer Research UK and is a trustee of Brain, PSP association and Darwin College.

**Patient and public involvement** Patients and/or the public were involved in the design, or conduct, or reporting, or dissemination plans of this research. Refer to the Methods section for further details.

**Patient consent for publication** Not applicable.

**Provenance and peer review** Not commissioned; externally peer reviewed.

**ORCID iDs**
Juliette Helene Lanskey http://orcid.org/0000-0002-6626-7686
Ece Kocagoncu http://orcid.org/0000-0002-6292-7472
Andrew J Quinn http://orcid.org/0000-0003-2267-9897
Jemma Pitt http://orcid.org/0000-0001-6256-6091
Vanessa Raymont http://orcid.org/0000-0001-8238-4279
Krish D Singh http://orcid.org/0000-0002-3094-2475
Mark Woolrich http://orcid.org/0000-0001-8460-8854
Anna C Nobre http://orcid.org/0000-0001-5762-2802
Richard N Henson http://orcid.org/0000-0002-0712-2639
James B Rowe http://orcid.org/0000-0001-7216-8679

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
