## [Reviewer comments · BMJ Open]

ARTICLE DETAILS

TITLE (PROVISIONAL)	The New Therapeutics in Alzheimer's Disease Longitudinal Cohort study (NTAD): study protocol
AUTHORS	Lanskey, Juliette; Kocagoncu, Ece; Quinn, Andrew; Cheng, Yun-Ju; Karadag, Melek; Pitt, Jemma; Isaac, John; Lowe, Stephen; Perkinson, Michael; Raymont, Vanessa; Singh, Krish; Woolrich, Mark; Nobre, Anna; Henson, Richard; Rowe, James

VERSION 1 – REVIEW

REVIEWER	Lyu, Jihui Beijing Geriatric Hospital, Center for Cognitive Disorders
REVIEW RETURNED	10-Sep-2021

GENERAL COMMENTS	This study aims to conduct a biomarker set for MEG /EEG for early detection and monitoring progression of AD. It is a meaningful research with reasonable design. However, there are still some minor problems and suggestions. 1. It would be more interesting and persuasive if there was a discussion section. If possible, a discussion deeply explaining the mechanism and possible results should be added.2. I am confused by one of the inclusion criteria for controls "Known amyloid status (CSF or PET). What does it mean and why?"3. In the exclusion criteria there is "Inability to walk 10 metres independently". Does this walking problem have any influence to the study?
---

REVIEWER	Smith, David University of Notre Dame, Psychology
REVIEW RETURNED	30-Sep-2021

GENERAL COMMENTS	BMJ-Open manuscript number bmjopen-2021-055135, entitled "The New Therapeutics in Alzheimer's Disease Longitudinal Cohort study (NTAD): study protocol", describes a protocol for ongoing research that is currently active at both investigational sites. As an ongoing investigation, there are as yet no Results to report or to review. This manuscript reports an effort to improve research methods in support of the search for new therapeutics for Alzheimer's disease processes. Among the primary study measures are MEG, EEG, and MRI, as well as a battery of neuropsychological and clinical tests, generally taken on two occasions over the course of one and two years. The longitudinal feature permits examination of disease progression. The sample includes 100 people with Alzheimer's disease (amyloid-positive mild impairment or early-stage disease) and 30 healthy controls (aged 50-85 years). 30 of the Alzheimer's patients will be retested 2 weeks after baseline for purposes of
--

estimating the reliability of MEG measures. Among analyses of change and psychometrics are ones designed to inform the design of clinical trials, by estimating power in relation to sample sizes, effect sizes, and measurement error.

There are a number of features of the proposed (ongoing) investigation that weaken its inferential prospects. In particular, the sample size seems small in relation to the proposed analyses, and most (all?) steps from accrual to the screenings and assessments are done by single individuals, making diagnostic reliability (e.g., psychiatric and some of the medical diagnoses) unknown.

There are also ambiguities in the description that leave wide latitude for interpretation. Specifically,

1. The Analysis section is quite underdeveloped. For instance, the page 7 statement, "Stage 1 analyses will assess the sensitivity of the cross-sectional MEG parameters to group effects (using parametric or non-parametric frequentist tests and receiver operating characteristic analyses and their Bayesian analogues), their correlation to baseline cognition (using Pearson or Spearman's rank correlation coefficient analyses) and their test-retest reliability (using intraclass correlation coefficient analyses)" does not restrict the analyses in any way whatsoever. Similarly, for Stage 2 the authors state they will use linear mixed models, but they do not describe any of these models. And the proposal to use multiple regression models to identify biomarkers predictive of change does not preclude the use of unsound approaches.

2. Figure 1, section A is not described sufficiently in the text, leaving important ambiguities, such as how those receiving a screening call are selected from those who are "invited", and what are the "biomarker status" screens for study eligibility? Similarly, page 8's "Potential participants are identified using local registry data, regional memory clinics and Join Dementia Research. People completing other observational studies may also be invited to screening" is quite vague, as is the nature of "candidates" who receive participant information sheets.

3. Page 9; How is the initial Alzheimer's Disease diagnosis made? Is this reliant on chart diagnoses? How are exclusion diagnoses made (e.g., psychiatric disorder)? What are the "clinically important abnormalities" or "clinically significant illnesses" that could compromise study participation? Who makes all these determinations and on what bases? Similarly (page 10), who does on-site screening? What "doctor" does clinical interview (e.g., clinical or research staff?), and what are the qualifications of "a member of the research team" who does other clinical assessments? Is it true that each step in the screening process is done by one individual, and therefore there will be no way to estimate the reliability of those judgments?

4. The authors' power analysis is based on the expectation of a group-wise effect size > 0.5 , which could be unrealistically large. Apart from that, the more sophisticated planned multivariate analyses reduce power even more. Relying on my interpretation of the authors' own estimates, it appears likely the study is under-powered, especially so for the more complex tests, such as correlates of disease progression over time. $N = 100$ seems a rather arbitrary number, and so absent further explanation of where the

	effect size estimates come from the power analyses appear to have been solved for an effect size that makes 100 sufficient. 5. The purpose of collecting DNA is not clear.
--	--

VERSION 1 – AUTHOR RESPONSE

Reviewer: 1

Dr. Jihui Lyu, Beijing Geriatric Hospital

Comments to the Author:

This study aims to conduct a biomarker set for MEG /EEG for early detection and monitoring progression of AD. It is a meaningful research with reasonable design. However, there are still some minor problems and suggestions.

1. It would be more interesting and persuasive if there was a discussion section. If possible, a discussion deeply explaining the mechanism and possible results should be added.

REPLY: Thank you. The journal's author guidelines on protocol papers do not include a discussion section. Recently published protocol papers have also not had discussions and we note the editors request to remove the conclusion section. Unless the editor suggests otherwise, we propose to leave the study discussion to the subsequent research papers reporting the outcomes of the study.

2. I am confused by one of the inclusion criteria for controls "Known amyloid status (CSF or PET). What does it mean and why?

REPLY: We apologise for the confusion. Whereas patient participants must be positive for Alzheimer biomarkers (in addition to their clinical diagnosis), the control subjects must have known biomarker status (but either positive or as is more common, negative). In other words, the controls may be biomarker negative (by cerebrospinal fluid or PET biomarkers), or they may be biomarker positive by virtue of age-related changes but as yet asymptomatic pathology. We clarify this in the revised manuscript (see section 'Overview of protocol stages – screening', page 8).

3. In the exclusion criteria there is "Inability to walk 10 metres independently". Does this walking problem have any influence to the study?

REPLY: Given the nature of mild cognitive impairment and Alzheimer's disease, mobility issues are not a common feature of the illness but may occur for other reasons in older adults. For practical reasons of using the MEG suite, participants need to have sufficient mobility to walk into the MEG suite unaided. We explain this in the revised manuscript (see Table 1 legend, page 7). In reality, we do not anticipate any significant impact on recruitment or bias in the type of mild cognitive impairment and Alzheimer's disease patient included in the study.

Reviewer: 2

Dr. David Smith, University of Notre Dame

Comments to the Author:

BMJ-Open manuscript number bmjopen-2021-055135, entitled "The New Therapeutics in Alzheimer's Disease Longitudinal Cohort study (NTAD): study protocol", describes a protocol for ongoing research that is currently active at both investigational sites. As an ongoing investigation, there are as yet no Results to report or to review.

This manuscript reports an effort to improve research methods in support of the search for new therapeutics for Alzheimer's disease processes. Among the primary study measures are MEG, EEG, and MRI, as well as a battery of neuropsychological and clinical tests, generally taken on two occasions over the course of one and two years. The longitudinal feature permits examination of disease progression. The sample includes 100 people with Alzheimer's disease (amyloid-positive mild impairment or early-stage disease) and 30 healthy controls (aged 50-85 years). 30 of the Alzheimer's patients will be retested 2 weeks after baseline for purposes of estimating the reliability of MEG measures. Among analyses of change and psychometrics are ones designed to inform the design of clinical trials, by estimating power in relation to sample sizes, effect sizes, and measurement error.

There are a number of features of the proposed (ongoing) investigation that weaken its inferential prospects. In particular, the sample size seems small in relation to the proposed analyses, and most (all?) steps from accrual to the screenings and assessments are done by single individuals, making diagnostic reliability (e.g., psychiatric and some of the medical diagnoses) unknown.

There are also ambiguities in the description that leave wide latitude for interpretation. Specifically,

1. The Analysis section is quite underdeveloped. For instance, the page 7 statement, "Stage 1 analyses will assess the sensitivity of the cross-sectional MEG parameters to group effects (using parametric or non-parametric frequentist tests and receiver operating characteristic analyses and their Bayesian analogues), their correlation to baseline cognition (using Pearson or Spearman's rank correlation coefficient analyses) and their test-retest reliability (using intraclass correlation coefficient analyses)" does not restrict the analyses in any way whatsoever. Similarly, for Stage 2 the authors state they will use linear mixed models, but they do not describe any of these models. And the proposal to use multiple regression models to identify biomarkers predictive of change does not preclude the use of unsound approaches.

REPLY: The protocol paper is not intended as a substitute for pre-registration of analyses, but to report the study design and data. It is anticipated that a data set of this complexity and multi-modality will be subject to different analytical approaches (data driven and model-based). By "unsound approaches" the reviewer might mean "p-hacking", but this is not the intention of the flexibility in this analytical overview.

Instead, the analysis section is intended as a high-level overview of the approaches to be taken. The study partners have differential expertise in 'traditional' analyses of evoked responses that are applicable to each task (Henson *et al.*, 2016); to data-driven approaches, such as the Hidden-Markov modelling of connectivity states (Baker *et al.*, 2014; Vidaurre *et al.*, 2016); and biophysical modelling of cortical dynamics (Adams *et al.*, 2020, 2021; Shaw *et al.*, 2020).

Such analyses can be preregistered, and we encourage analysts to do so, within the consortium (e.g. we will have predefined statistical analysis strategies) and after data dissemination. In the revised paper (pages 4 and 5), we highlight the approaches anticipated for analysis, in primary analyses of the data (hypotheses testing) and the scope for exploratory analysis (hypothesis-generating). The combination of NTAD data with other data sets, such as BIOFIND or ENIGMA, is beyond the scope of the current protocol but would of course enable cross-validation of results and hypothesis generation of hypothesis testing.

2a. Figure 1, section A is not described sufficiently in the text, leaving important ambiguities, such as how those receiving a screening call are selected from those who are "invited"

REPLY: Thank you. We have included clarification in the revised manuscript in the 'Participant recruitment and selection' method section (pages 5 and 6).

Participants are selected for stage 1 based on whether they meet the inclusion and exclusion criteria (Table 1) following screening of (1) electronic records, (2) telephone responses, (3) on-site review and (4) biomarker results.

2b. And what are the "biomarker status" screens for study eligibility?

REPLY: The biomarker status is determined using PET or CSF. In the 'overview of protocol status – screening' section (page 8), we have changed 'amyloid status' to 'Alzheimer biomarker status' for clarity. Further details about CSF and PET procedures are under their respective headings (pages 9 and 10).

2c. Similarly, page 8's "Potential participants are identified using local registry data, regional memory clinics and Join Dementia Research. People completing other observational studies may also be invited to screening" is quite vague, as is the nature of "candidates" who receive participant information sheets.

REPLY: We realise this was ambiguous. The screening stage has 3 principal steps:

- ➔ electronic screening of medical/database records at the register (held by memory clinic or Join Dementia Research)
(if eligible, they are sent the participant information sheet)
- ➔ telephone based questionnaire
(if still eligible, invited to on-site screening)
- ➔ on-site screening including PET or CSF
(if still eligible, move to stage 1 of study)

Included in the local registry are people who have expressed interest in hearing about further research opportunities, some of whom had previously participated in other observational studies. They follow the same screening procedures. We have re-written the section to explain this process more clearly (see section 'participant selection and recruitment', pages 5 and 6; and 'Overview of protocol stages – screening', page 8).

3a. Page 9; How is the initial Alzheimer's Disease diagnosis made?

REPLY: The initial diagnosis is made at a memory clinic prior to our study. Confirmation of the diagnosis of MCI or AD is made during the study based on medical records and in-study cognitive and biomarker assessment, according to Albert *et al.* (2011) and McKhann *et al.* (2011) with clinical, imaging and biomarker evidence.

We have updated the manuscript ('participant selection and recruitment' section, pages 5 and 6) to clarify where the AD/MCI diagnosis is made.

3b. Is this reliant on chart diagnoses?

REPLY: The initial diagnosis from which participants are invited to screening is reliant on NHS memory assessment services. Clinical records are available to the study team, and these data are supplemented by assessment of clinical diagnostic criteria and biomarkers, by the study team as clarified in the revised manuscript.

3c. How are exclusion diagnoses made (e.g., psychiatric disorder)? What are the "clinically important abnormalities" or "clinically significant illnesses" that could compromise study participation? Who makes all these determinations and on what bases?

REPLY: The researcher screening participants checks the participant medical records and clinical letters to ensure they do not have additional diagnoses from the exclusion criteria. During the phone screening, participants are asked about other diagnoses:

*“Have you been diagnosed with a neurological disorder?
Have you been diagnosed with a psychiatric disorder or any other clinical condition?
Have you had a head trauma that resulted in loss of consciousness in the last 5 years?
Are you aware of an infection in your body (e.g. HIV, hepatitis B or C)?”
Do you have a history of epileptic seizures?*

During the clinical interview, the participant and their caregiver are asked by a medically qualified doctor to give any relevant medical history from the following systems: cardiovascular, respiratory, hepato-biliary, gastrointestinal, genitourinary, endocrine, haematological, Musculo-skeletal, neoplasia, neurological, psychological, immunological, dermatological, allergies, eyes, ears, nose, throat, other.

They are also asked “in the past 10 years has the subject ever been admitted to a hospital for more than two days for any disorder or condition? If yes, specify reason and list hospital admissions with approximate dates.”

Where a disorder is present but of uncertain severity or relevance, this is discussed with the PI for the relevant site. As examples, a history of epilepsy would be exclusionary but mild migraine would not; bipolar disorder would be exclusionary but a history of remitted mild anxiety or depression would not.

We have clarified this in the ‘participant recruitment and selection’ section (pages 5 and 6).

3d. Similarly (page 10), who does on-site screening?

REPLY: On site screening is carried out by a member of the research team who may or may not also be a clinician in the memory assessment service, as detailed in the ‘screening’ section (page 8).

3e. What "doctor" does clinical interview (e.g., clinical or research staff)?

REPLY: A medically qualified doctor, with at least MRCP part II qualifications and in ST3+ level training, or completion of neurology or psychiatry specialist training, including cognitive disorders experience. We have clarified in the revised manuscript (page 8).

For international comparison, the ST3+ specialty training approximates to a neurology/psychiatry resident beginning 3-4 years after completion of a doctor’s primary medical degree. MRCP is membership of the royal college of physicians, a training examination on completion of core medical training (cf. internal medicine) prior to beginning ST3+ specialty training (cf. residency).

3f. and what are the qualifications of "a member of the research team" who does other clinical assessments?

REPLY: All members of the research team have current Good Clinical Practice (GCP) training.

Those undertaking clinical interviews and assessments (including clinical interview and Clinical Dementia Rating) are medically qualified with at least MRCP part II qualifications and are either in specialty training or have completed training (CCT) in neurology or psychiatry, including cognitive disorders experience.

We have specified this in the revised manuscript (page 8).

3g. Is it true that each step in the screening process is done by one individual, and therefore there will be no way to estimate the reliability of those judgments?

REPLY: CSF Biomarker assays for screening are performed by UCLH. For amyloid PET scans, they are reported by a qualified reporter outside of the study team at either University of Cambridge or University of Oxford. Later, estimation of the centiloid is calculated by a member of the research team. The CSF and/or PET results are discussed with a study doctor to confirm eligibility based on standardised threshold criteria

The clinical interview and Clinical Dementia Rating are performed by a medically qualified doctor as above.

The MMSE may be performed and scored by other members of the research team where they have experience in the test, and not necessarily one individual.

Exclusion or inclusion criteria are checked off the checklist in the protocol. The medical notes and clinical letters are reviewed by a member of the research team to complete the checklist. If there is any additional information from this review which is to inform either inclusion or exclusion, this is checked with the PI at each site. We have clarified this in the participant selection and recruitment section. See pages 6, 8 and 10.

4. The authors' power analysis is based on the expectation of a group-wise effect size > 0.5 , which could be unrealistically large. Apart from that, the more sophisticated planned multivariate analyses reduce power even more. Relying on my interpretation of the authors' own estimates, it appears likely the study is under-powered, especially so for the more complex tests, such as correlates of disease progression over time. $N = 100$ seems a rather arbitrary number, and so absent further explanation of where the effect size estimates come from the power analyses appear to have been solved for an effect size that makes 100 sufficient.

REPLY: We do not agree with the assertion that the study is underpowered, for the following reasons.

Empirically, it is clear from previous published studies that some imaging markers can detect the presence of MCI/AD and the annual rate of change with as few as 50 participants. For example, for several MRI metrics in ADNI (Holland *et al.* 2009, see their figure 2 inset below); and similar sensitivity for measures derived from MEG (Fernández *et al.*, 2006; Pusil *et al.*, 2019; Xu *et al.*, 2021) and EEG (Papaliagkas *et al.*, 2008, 2011; Lai *et al.*, 2010; Bennys *et al.*, 2011; Chapman *et al.*, 2011; Fruehwirt *et al.*, 2019). Given the sensitivity of MRI as an already common tool in early phase clinical trials, one of the aspirations of NTAD is to identify MEG measures that are at least as sensitive as MRI. Indeed, current longitudinal E/MEG studies show significant effects with our planned sample size or less (see below tables).

With a more formal approach to power, we have calculated that we are adequately powered for a medium effect size ($d \sim 0.5$) for the group-wise comparison. Please see Table 1 of Mandal and colleagues, 2018 for a summary of studies comparing people with Alzheimer's disease to controls and the resultant area under the curves. 30/34 MEG measures were AUC > 0.64 approximating Cohen's $d > 0.5$ (Salgado, 2018).

For the power of longitudinal analyses, our study is also adequately powered for detecting the correlates of disease progression for *smaller* effect sizes > 0.28 , even with 20% attrition. For example, power analyses using Gpower3 indicate the effect size for which our sample size has 80% power; from which we can assess the plausibility of the effect size from the literature. For MEG measures to

be adopted alongside or even instead of MRI measures, we anticipate the need to be more sensitive to disease effects. We have clarified the estimation and planning of power in the revised manuscript (page 8).

Previous neurophysiological studies with similar sample sizes have enough power

EEG

Finding	Reference	Sample size	Duration
Significant differences between baseline and follow-up EEG	(Bennys et al. , 2011)	71 MCI	12 months
	(Lai et al. , 2010)	20 AD 18 MCI	12 months
	(Fruehwirt et al. , 2019)	63 AD	18 months
	(Papaliagkas et al. , 2011)	22 MCI	14 & 23 months
Baseline EEG scan predictive of conversion from MCI to AD	(Chapman et al. , 2011)	43 MCI	11 months
	(Papaliagkas et al. , 2008)	91 MCI, 30 HC	14 months

MEG

Finding	Reference	Sample	Duration
Significant differences between baseline and follow-up MEG	(Pusil et al. , 2019)	54 MCI	24 months
Baseline MEG scan predictive of conversion from MCI to AD	(Fernández et al. , 2006)	19 AD, 17 MCI, 17 HC	24 months
	(Xu et al. , 2021)	76 MCI, 53 HC	36 months

Open in a separate window

Fig. 2.

Sample size estimates for AD from a linear mixed-effects model with random slopes. The bars, with 95% confidence intervals, indicate the expected number of subjects needed per arm to detect a 25% reduction in rate of change at the $P < 0.05$ level with 80% power, assuming a 24-month trial with scans every 6 months. Results for Model T are in blue and results for Model D are in red; numerical values are shown in Table 1.

Our effect sizes were solved using standard techniques in G*power

5. The purpose of collecting DNA is not clear.

REPLY: There are several reasons for collecting DNA. First is estimation of APOE status. Second is to contribute to genotyping studies of MCI/AD in collaboration with external multicentre studies and compare our participants with larger cohort studies in the UK. Third is to conduct exploratory post hoc analysis of any potential association between autosomal dominant cases and distinctive physiology and phenotype. Fourth is to be in a position to test hypotheses emerging in relation to common polymorphisms of genes that regulate synaptic function or cognition or Alzheimer risk. These are exploratory genetic options enabled by the availability of DNA, while participation and the principal analyses are not genetically determined. We have included these reasons within the revised manuscript (Analysis section, page 5).

References

Adams NE, Hughes LE, Phillips HN, Shaw AD, Murley AG, Nesbitt D, et al. GABA-ergic dynamics in human frontotemporal networks confirmed by pharmaco-magnetoencephalography. *J Neurosci* 2020; 1689–19.

Adams NE, Hughes LE, Rouse MA, Phillips HN, Shaw AD, Murley AG, et al. GABAergic cortical network physiology in frontotemporal lobar degeneration. *Brain* 2021; 144: 2135–45.

Albert MS, DeKosky ST, Dickson D, Dubois B, Feldman HH, Fox NC, et al. The diagnosis of mild cognitive impairment due to Alzheimer's disease: Recommendations from the National Institute on Aging-Alzheimer's Association workgroups on diagnostic guidelines for Alzheimer's disease. *Alzheimers Dement* 2011; 7: 270.

Baker AP, Brookes MJ, Rezek IA, Smith SM, Behrens T, Smith PJP, et al. Fast transient networks in spontaneous human brain activity. *Elife* 2014; 2014

Bennys K, Rondouin G, Benattar E, Gabelle A, Touchon J. Can event-related potential predict the progression of mild cognitive impairment? *J Clin Neurophysiol* 2011; 28: 625–32.

Chapman RM, McCrary JW, Gardner MN, Sandoval TC, Guillily MD, Reilly LA, et al. Brain ERP components predict which individuals progress to Alzheimer's disease and which do not. *Neurobiol Aging* 2011; 32: 1742–55.

Fernández A, Turrero A, Zuluaga P, Gil P, Maestú F, Campo P, et al. Magnetoencephalographic Parietal δ Dipole Density in Mild Cognitive Impairment: Preliminary Results of a Method to Estimate the Risk of Developing Alzheimer Disease. *Arch Neurol* 2006; 63: 427–30.

Fruehwirt W, Dorffner G, Roberts S, Gerstgrasser M, Grossegger D, Schmidt R, et al. Associations of event-related brain potentials and Alzheimer's disease severity: A longitudinal study. *Prog Neuropsychopharmacol Biol Psychiatry* 2019; 92: 31–8.

Henson RN, Campbell KL, Davis SW, Taylor JR, Emery T, Erzinclioglu S, et al. Multiple determinants of lifespan memory differences. *Sci Reports* 2016 61 2016; 6: 1–14.

Holland D, Brewer JB, Hagler DJ, Fennema-Notestine C, Dale AM, Initiative the ADN, et al. Subregional neuroanatomical change as a biomarker for Alzheimer's disease. *Proc Natl Acad Sci U S A* 2009; 106: 20954.

Lai CL, Lin RT, Liou LM, Liu CK. The role of event-related potentials in cognitive decline in Alzheimer's disease. *Clin Neurophysiol* 2010; 121: 194–9.

Mandal PK, Banerjee A, Tripathi M, Sharma A. A Comprehensive Review of Magnetoencephalography (MEG) Studies for Brain Functionality in Healthy Aging and Alzheimer's Disease (AD). *Front Comput Neurosci* 2018; 12: 60.

McKhann GM, Knopman DS, Chertkow H, Hyman BT, Jack CR, Kawas CH, et al. The diagnosis of dementia due to Alzheimer's disease: Recommendations from the National Institute on Aging-Alzheimer's Association workgroups on diagnostic guidelines for Alzheimer's disease. *Alzheimer's Dement* 2011; 7: 263–9.

Papaliagkas V, Kimiskidis V, Tsolaki M, Anogianakis G. Usefulness of event-related potentials in the assessment of mild cognitive impairment. *BMC Neurosci* 2008; 9: 1–10.

Papaliagkas VT, Kimiskidis VK, Tsolaki MN, Anogianakis G. Cognitive event-related potentials: Longitudinal changes in mild cognitive impairment. *Clin Neurophysiol* 2011; 122: 1322–6.

Pusil S, López ME, Cuesta P, Bruña R, Pereda E, Maestú F. Hypersynchronization in mild cognitive impairment: the 'X' model. *Brain* 2019; 142: 3936–50.

Salgado JF. Transforming the area under the normal curve (AUC) into cohen's d, pearson's rpb, odds-ratio, and natural log odds-ratio: Two conversion tables. *Eur J Psychol Appl to Leg Context* 2018; 10: 35–47.

Shaw AD, Muthukumaraswamy SD, Saxena N, Sumner RL, Adams NE, Moran RJ, et al. Generative modelling of the thalamo-cortical circuit mechanisms underlying the neurophysiological effects of ketamine. *Neuroimage* 2020; 221: 117189.

Vidaurre D, Quinn AJ, Baker AP, Dupret D, Tejero-Cantero A, Woolrich MW. Spectrally resolved fast transient brain states in electrophysiological data. *Neuroimage* 2016; 126: 81–95.

Xu M, Sanz DL, Garces P, Maestu F, Li Q, Pantazis Di. A Graph Gaussian Embedding Method for Predicting Alzheimer's Disease Progression with MEG Brain Networks. *IEEE Trans Biomed Eng* 2021; 68: 1579–88.

VERSION 2 – REVIEW

REVIEWER	Lyu, Jihui Beijing Geriatric Hospital, Center for Cognitive Disorders
REVIEW RETURNED	29-Dec-2021

GENERAL COMMENTS	I think all my questions have been considered and in the revised manuscript.
--

REVIEWER	Smith, David University of Notre Dame, Psychology
REVIEW RETURNED	23-Jan-2022

GENERAL COMMENTS	BMJ-Open manuscript number bmjopen-2021-055135R, entitled "The New Therapeutics in Alzheimer's Disease Longitudinal Cohort study (NTAD): Study protocol", is a revised version of a protocol description for ongoing research designed to improve research methods in support of the search for new therapeutics for Alzheimer's disease processes. 1. The authors responded to my prior request for data-analytic details by stating that protocol papers "report the study design and data", and that analyses are to be pre-registered elsewhere. In contrast, my original review was guided by reviewer instructions that state, "Publishing protocols in full also makes available more
--

	information than is currently required by trial registries and increases transparency, making it easier for others (editors, reviewers and readers) to see and understand any deviations from the protocol that occur during the conduct of the study." I leave it to the Editor to decide whether my original requests were appropriate and note only that without such details I cannot evaluate the adequacy of the statistical analyses. 2. My prior remark about unsound statistical approaches did not refer to p-hacking; it was to suggest that there was insufficient description of the "multiple regression models" to evaluate whether they capture the effects being targeted. 3. The power analysis remains elusive, inasmuch as power is ordinarily in relation to a specific analysis. The G*power plot is for a one-tailed paired-sample t-test, and it shows that to achieve the desired power of .80 with an effect size ~.50 the authors would need ~25 participants. From this plot they conclude that 100 participants would permit detection of much smaller effect sizes. This is of course true, but it is quite imprecise to then imagine on this plot the more complex multivariate situations implied by their design description and study ambitions (e.g., incremental contribution of biomarker sets, longitudinal mediation modeling), especially once variables of widely varying precision (e.g., individual spectral density and connectivity scores) start entering into the models. If this plot is to be published, the more detailed one from the author response letter should be used instead of the one currently in the manuscript so that readers know to what analysis it applies. 4. The authors note in their response to my review point #1 that "The study partners have differential expertise [in various analyses]." The manuscript further notes ("Analysis" paragraph 1) that "investigators" will identify harmonized pipelines, that investigators will address the study aims with multiple analytical approaches, and that investigators ("analysts" in the response letter) are encouraged to pre-register their analyses. This is a bit puzzling as it suggests these partners/ analysts/ investigators are not authors on this manuscript. If the investigators are indeed authors, then wouldn't this section state, for instance, that specific analyses WILL be pre-registered? And wouldn't expertise be the same shared expertise of the collaborating authors? Or, maybe this "protocol" is limited to harmonizing data collection and building a data set, leaving specific studies of these data up to partner and investigator initiatives at later times? In conclusion, if my fussiness about some statistical details is inappropriate at this stage, then I am comfortable moving this manuscript forward, especially if the text can be modified to declare the purpose of the study to be assuring a strong data set emerges from the ambitious data collection effort currently underway, leaving what is done with those data to future reports and future peer review.
--	--

VERSION 2 – AUTHOR RESPONSE

Reviewer: 1
Dr. Jihui Lyu, Beijing Geriatric Hospital

Comments to the Author:
I think all my questions have been considered and in the revised manuscript.

Reviewer: 2
Dr. David Smith, University of Notre Dame

Comments to the Author:
BMJ-Open manuscript number bmjopen-2021-055135R, entitled "The New Therapeutics in Alzheimer's Disease Longitudinal Cohort study (NTAD): Study protocol", is a revised version of a protocol description for ongoing research designed to improve research methods in support of the search for new therapeutics for Alzheimer's disease processes.

1. The authors responded to my prior request for data-analytic details by stating that protocol papers "report the study design and data", and that analyses are to be pre-registered elsewhere. In contrast, my original review was guided by reviewer instructions that state, "Publishing protocols in full also makes available more information than is currently required by trial registries and increases transparency, making it easier for others (editors, reviewers and readers) to see and understand any deviations from the protocol that occur during the conduct of the study." I leave it to the Editor to decide whether my original requests were appropriate and note only that without such details I cannot evaluate the adequacy of the statistical analyses.

Thank you for your comments. We understand the need to clarify the purpose and scope of this protocol paper. The primary aim is to detail the conduct of the data collection in this large, multi-site study and the nature of the data that will be available for future analytical studies. Because of the large and open-ended nature of the analyses afforded by the data set, it is not possible to describe them exhaustively and in detail.

We have clarified in the manuscript that future, separate, analytical papers will need to detail their full methods in peer-reviewed journals (see pages 4-5). We recommend pre-registration of analysis but do not enforce it, and there remains a place for exploratory analysis and the assessment of new analysis tools, provided this is made clear in a report. Moreover, some methods to be applied to the dataset in due course are not routine to a degree that their details could yet be set out as a pre-registration in this protocol paper. Optimisation or validation of such new methods will need to be completed before application to the NTAD data, but cannot be set out in full now. As the reviewer correctly summarises later, this protocol focusses on the issues of harmonizing data collection and building a data set, leaving specific studies of these data to investigator initiatives.

We note the editors request that we "signpost to where the information is available or will be available at a later date/future plans." We have updated the manuscript to signpost that analytical papers with full methods will be published in the future which cite this protocol paper on pages 4-5.

2. My prior remark about unsound statistical approaches did not refer to p-hacking; it was to suggest that there was insufficient description of the "multiple regression models" to evaluate whether they capture the effects being targeted.

We appreciate the confusion. Detailed prior specification of the multiple regression models is best undertaken in a pre-registration report, and is beyond the scope of this protocol paper. We have

revised the manuscript on page 5 to address this.

3. The power analysis remains elusive, inasmuch as power is ordinarily in relation to a specific analysis. The G*power plot is for a one-tailed paired-sample t-test, and it shows that to achieve the desired power of .80 with an effect size $\sim .50$ the authors would need ~ 25 participants. From this plot they conclude that 100 participants would permit detection of much smaller effect sizes. This is of course true, but it is quite imprecise to then imagine on this plot the more complex multivariate situations implied by their design description and study ambitions (e.g., incremental contribution of biomarker sets, longitudinal mediation modelling), especially once variables of widely varying precision (e.g., individual spectral density and connectivity scores) start entering into the models. If this plot is to be published, the more detailed one from the author response letter should be used instead of the one currently in the manuscript so that readers know to what analysis it applies.

Thank you. We understand this concern and agree that power will be dependent on the specific analysis and variables in question and have updated the manuscript on page 7 to reflect this. This is one reason why a given analysis plan would benefit from a pre-registration report, which would be able to provide the specific power estimates for the analysis outlined. Here we provided indicative power estimates for common simple tests, to contextualise the power for given effects sizes that arise from a study of this scale. We have thus revised figure 2 and its legend (see page 17) to include the supplementary values used in the calculation, i.e. that effect sizes were estimated using G*power3.1 for one-tailed, paired t-tests using 80% power and $\alpha=0.05$. This figure is we believe a useful reference where the prior literature can indicate the expected effect size of AD on different types of outcome measure (cognitive, MRI volume, M/EEG etc). It is not of course a substitute for a bespoke power calculation for more complex methods, including multivariate analysis.

4. The authors note in their response to my review point #1 that "The study partners have differential expertise [in various analyses]." The manuscript further notes ("Analysis" paragraph 1) that "investigators" will identify harmonized pipelines, that investigators will address the study aims with multiple analytical approaches, and that investigators ("analysts" in the response letter) are encouraged to pre-register their analyses. This is a bit puzzling as it suggests these partners/analysts/ investigators are not authors on this manuscript. If the investigators are indeed authors, then wouldn't this section state, for instance, that specific analyses WILL be pre-registered? And wouldn't expertise be the same shared expertise of the collaborating authors? Or, maybe this "protocol" is limited to harmonizing data collection and building a data set, leaving specific studies of these data up to partner and investigator initiatives at later times?

We apologise for the confusion. The priority for this protocol paper is to describe the study design, data collections, and harmonisation of data collection across sites as a reference for future analytical studies with in-depth detail of how this dataset is being built (updated in manuscript on pages 4-5). We anticipate the dataset being widely used, by many investigators outside of the protocol paper authorship. We recommend pre-registration, and authors of future work should be honest and clear about exploratory versus planned analysis, but we do not feel it appropriate for a protocol paper to mandate pre-registration, nor for this protocol paper to serve as pre-registration document.

For example, there may be an analytical method X that is already anticipated, but which need to be optimised or validated using other data before it would be applied to NTAD data. Investigators may already have a broad plan of analysis using method X, but not be able yet to commit to the pre-registration because the full details of the application of X cannot yet be stated. After the method X is optimised and validated, the investigators would be able to submit a pre-registration report to use method X on the NTAD data. Clearly it would be inappropriate to use the NTAD data for the optimisation of X and then re-apply X to the same data.

5. In conclusion, if my fussiness about some statistical details is inappropriate at this stage, then I am comfortable moving this manuscript forward, especially if the text can be modified to declare the purpose of the study to be assuring a strong data set emerges from the ambitious data collection effort currently underway, leaving what is done with those data to future reports and future peer review.

Thank you for your comments and summary. The main aim of this protocol paper is to describe the extensive data collection underway, with harmonisation across sites, and to allow future analytical reports to this protocol of data acquisition and scope. Those papers will need to describe their individual research methods in detail, whether hypothesis testing or exploratory, pre-registered or not, with bespoke power calculations for their univariate or multivariate methods, whether at baseline or in longitudinal modelling. We have updated the manuscript to reflect this, see page 5.

Reviewer: 1

Competing interests of Reviewer: No

Reviewer: 2

Competing interests of Reviewer: none

***Comment from the editorial assistant: Thank you for submitting your manuscript entitled "The New Therapeutics in Alzheimer's Disease Longitudinal Cohort study (NTAD): study protocol" (manuscript ID bmjopen-2021-055135.R2) to BMJ Open. This has been returned to you to address the following issues before it can be assigned to the Editor.

1. Funding Information (award/grant number). You have indicated a funder/s for your paper. Please ensure to provide an award/grant number for your funder/s in the main document file and in ScholarOne. Please put 'no award/grant number' if necessary.

The study is primarily funded by DPUK, for which we give the award number. For transparency, we declare the background funders to DPUK. There is no number system applied to these background funders and the appropriate number is that of the DPUK award to the study team. We have updated the manuscript and ScholarOne to reflect this.

2. Please ensure that all and only co-authors of this submission must be included/mentioned in contributorship statement.

We have replaced 'all authors' with each author's initials for clarity in the contributorship statement.

3. Please confirm if the uploaded figure 4 is downloaded from the internet as we will need you to provide a permission from the publishers for it. If it was illustrated by one of the co-authors, or it was commissioned from the artist, you will still need to provide a permission and proper declaration of it in your figure legend.

The overall figure was created by Juliette Lanskey and the photos in 4c have been updated to illustrative images owned by Juliette Lanskey; I give permission for this to be published. The icons for (d) were designed by Freepik and the legend has been updated to meet the Freepik licence requirements.

4. Please cite references in the main text in numerical order *86-87

We have switched 86 and 87.

VERSION 3 – REVIEW

REVIEWER	Smith, David University of Notre Dame, Psychology
REVIEW RETURNED	12-Oct-2022

GENERAL COMMENTS	BMJ-Open manuscript number bmjopen-2021-055135.R2, entitled "The New Therapeutics in Alzheimer's Disease Longitudinal Cohort study (NTAD): Study protocol", is the second revision of a protocol description for ongoing research designed to improve research methods in support of the search for new therapeutics for Alzheimer's disease processes. I was Reviewer 2 on the prior submission. My main concern with the manuscript has been the implicit mixing of data collection pre-registration ("protocol paper") and data analysis pre-registration. The authors are now quite explicit in characterizing this manuscript as being about how a data set will be created, with the analyses applied to those data being deferred to future investigators. Despite these explicit statements, which are described in the author response letter, the paper continues to include data analytic details that are not described sufficiently to permit evaluation (e.g., the power figure, longitudinal mediation testing). Such detail belies the newer explicit statements about this being a protocol paper. Nevertheless, with the many explicit qualifying statements now in place I am comfortable moving the manuscript forward.
---